# Full-Length Transcriptome Analysis of the Ichthyotoxic Harmful Alga *Heterosigma akashiwo* (Raphidophyceae) Using Single-Molecule Real-Time Sequencing

**DOI:** 10.3390/microorganisms11020389

**Published:** 2023-02-03

**Authors:** Nanjing Ji, Xueyao Yin, Yujiao Chen, Yifan Chen, Mingyang Xu, Jinwang Huang, Yuefeng Cai, Xin Shen

**Affiliations:** 1Jiangsu Key Laboratory of Marine Bioresources and Environment/Jiangsu Key Laboratory of Marine Biotechnology, Jiangsu Ocean University, Lianyungang 222005, China; 2Co-Innovation Center of Jiangsu Marine Bio-Industry Technology, Jiangsu Ocean University, Lianyungang 222005, China

**Keywords:** harmful alga, *Heterosigma akashiwo*, SMRT sequencing, reference transcriptome

## Abstract

The raphidophyte *Heterosigma akashiwo* is a harmful algal species. The bloom of this organism has been associated with the massive mortality of fish in many coastal waters. To investigate the molecular mechanism of *H*. *akashiwo* blooms, having a reliable reference transcriptome of this species is essential. Therefore, in this study, a full-length transcriptome of *H*. *akashiwo* was obtained by single-molecule real-time sequencing. In total, 45.44 Gb subread bases were generated, and 16,668 unigenes were obtained after the sequencing data processing. A total of 8666 (52.00%) unigenes were successfully annotated using seven public databases. Among them, mostly phosphorus and nitrogen metabolism genes were detected. Moreover, there were 300 putative transcription factors, 4392 putative long non-coding RNAs, and 7851 simple sequence repeats predicted. This study provides a valuable reference transcriptome for understanding how *H*. *akashiwo* blooms at a molecular level.

## 1. Introduction

Harmful algae blooms (HABs) occur when algal species grow out of control and produce toxic or harmful effects on the ecosystem. In recent decades, a considerable amount of evidence indicates that HABs have increased in frequency and intensity, causing serious impacts on fisheries, tourism, and even public health [1,2,3]. The marine microalgae *Heterosigma akashiwo* (Raphidophyceae) is a typical HAB species and has a widespread global distribution. This species is notorious because its blooms have caused the massive mortality of cultivated fish but the mechanism for its ichthyotoxicity is not well resolved and remains controversial [4,5]. For instance, in March-April 2021, a severe bloom of *H*. *akashiwo* occurred in the Comau fjord, causing more than 6000 t salmon mortalities [6]. In addition, the growth of *H*. *akashiwo* showed a significant increase in response to an elevated temperature and carbon dioxide (CO_2_) level when compared with the co-occurring dinoflagellate *Prorocentrum minimum* [7]. Additionally, CO_2_ enrichment also benefits the growth of *H*. *akashiwo* over the diatom *Skeletonema costatum* [8]. These studies have highlighted that the bloom of *H*. *akashiwo* may increase in severity under climate change pressures [3,9].

Many studies have been conducted to understand the bloom formation mechanisms of *H*. *akashiwo*. There are many environmental factors (e.g., light, temperature, and nutrients) involved in regulating *H*. *akashiwo* bloom formations [10,11,12]. For instance, previous studies have shown that *H*. *akashiwo* can utilize different nitrogen (N) sources (nitrate, ammonium, and urea) and dissolved organic phosphorus (e.g., glucose-6-phosphate and adenosine 5-triphosphate) [11,13,14,15]. Recently, Healey et al. (2023) proposed that utilization of nitric oxide (NO) may give *H*. *akashiwo* a competitive advantage over other phytoplankton in regions with high NO concentrations [16]. According to the physiological strategies of HAB species, including nutrient acquisition styles, behaviors, and biological interactions with coexisting species, Jeong et al. (2015) categorized the HAB dynamics into four generation mechanisms (GMs; GM1–GM4), and GM4 can be used to explain the bloom of *H*. *akashiwo* [17]. For instance, diurnal vertical migration behavior enables *H*. *akashiwo* to acquire nutrients (e.g., phosphate) from deeper layers and capture light in the surface layer [18,19]. In addition, previous studies found that *H*. *akashiwo* can feed on autotrophic and heterotrophic bacteria [20]. This indicates that *H*. *akashiwo* mixotrophy also probably affects the bloom of this species.

Recently, omics approaches (e.g., genomics, transcriptomics, and proteomics) have changed the landscape of many fields, including the study of HABs. They have provided a revolutionary tool for understanding the ecology of HAB species and bloom dynamics [21,22]. For instance, transcriptomics and gene expression profiling have frequently been used in HAB studies. In 2014, the Marine Eukaryotic Microbial Transcriptome Project published nearly 680 transcriptomes, and about 150 belong to HAB genera [22,23]. Among them, 14 treatment groups were designed for four different strains of *H*. *akashiwo*. For instance, the transcriptional responses of *H*. *akashiwo* to nutrient stress (nitrate and phosphate) were studied using next-generation sequencing (NGS) [13,24]. Furthermore, NGS was also used to explore the gene expression of the response of *H*. *akashiwo* to rising CO_2_ levels [9]. However, the fundamental limitation of NGS is short read lengths, which makes it difficult to accurately obtain complete transcripts due to the complex genome structure. In recent years, third-generation sequencing (e.g., single-molecule real-time [SMRT]) technologies have been able to address the shortcoming of NGS. Single-molecule real-time sequencing can obtain full-length transcript sequences, without assembly, providing a better opportunity to explore whole-transcriptome profiling in non-model species [25,26,27,28].

Consequently, in this study, a full-length transcriptome of *H*. *akashiwo* was generated using SMRT sequencing. Then, gene functional annotation, coding sequence (CDS) prediction, and transcription factor (TF) and long non-coding RNA (lncRNA) prediction, were conducted. In addition, the genes that are involved in several physiological functions, including phosphorus (P) and N metabolism, were selected for further discussion. The data presented in this study provides a full-length transcriptome for the study of the ecology and physiology of *H*. *akashiwo*.

## 2. Materials and Methods

### 2.1. Heterosigma Akashiwo Culture and Sampling

*Heterosigma akashiwo* (strain: CCMA369) was provided by the Center for Collection of Marine Algae at Xiamen University. This strain was originally isolated from the East China Sea (20 May 2011). *H*. *akashiwo* was cultured in f/2-Si medium, which was prepared with 0.22 μm filtered and autoclaved natural seawater (a salinity is approximately 30 psu). The culture was maintained at 20 ℃ under a 14:10 h light: dark cycle, with an average photo flux of 100 μE m^−2^ s^−1^. The cell concentration was determined by the cell count using a Sedgwick–Rafter chamber under a microscope.

To obtain abundant transcriptomic information for *H*. *akashiwo*, diel sampling was performed. Briefly, 12 flasks of *H*. *akashiwo* were inoculated at the same time and concentration. When the cultures were in the exponential growth stage (the concentration of cell is about 15 × 10^4^ cell/mL), the sampling began 2 h (S1, day 1, 22:00) after the dark period (D0, day1, 20:00), and, subsequently, the samples were collected at 10 (S2, day2, 6:00), 14 (S3, day2, 10:00), and 20 (S4, day2, 18:00) h after D0. All the samples (S1, S2, S3 and S4) in this study were set up with three biological replicates. To avoid the sampling process interfering with the cells, the samples in each culture flask were only collected once. The cells of *H*. *akashiwo* were collected by centrifugation at 3000 rpm for 5 min at 20 ℃. The cell pellets were transferred to a set of 1.5 mL microcentrifuge tubes. Then, 1 mL of TRIzol reagent (Molecular Research Center, Cincinnati, OH, USA) and approximately 0.2 g of 0.1 mm diameter ceramic beads were added into each sample tube. The samples were homogenized using a FastPrep-24 homogenizer at 6 m/s for 30 s, and then they were stored at −80 ℃ until RNA isolation.

### 2.2. RNA Isolation and Single-Molecule Real-Time Sequencing

The total RNA was isolated using the TRIzol method, and further purification was performed using Direct-zol RNA Miniprep (Zymo Research, Irvine, CA, USA) as was previously described [29]. The integrity of the RNA was initially monitored by agarose electrophoresis. Furthermore, the RNA purity and concentration were determined using a biophotometer (Eppendorf, Hamburg, Germany) and Qubit RNA Assay Kit with a Qubit 2.0 Fluorometer (Life Technologies, Carlsbad, CA, USA), respectively. Thereafter, the RNA integrity was further examined with an Agilent 2100 Bioanalyzer (Agilent Technologies, Palo Alto, CA, USA). Subsequently, an RNA integrity number ≥7 was considered for PacBio sequencing library construction.

An equal amount of RNA from each sample was mixed. The full-length complementary DNA (cDNA) was synthesized using a Clontech SMARTer PCR cDNA Synthesis Kit (Clontech, Mountain View, CA, USA) according to the manufacturer’s instructions. Then, polymerase chain reaction (PCR) amplification was performed on the cDNA. Next, size selection was conducted with BluePippin, and large-scale PCR was performed. Afterward, full-length cDNA damage/terminal repair was carried out, and SMRT bell libraries were constructed. The library sequencing was performed on the PacBio Sequel platform at Novogene Co. Ltd. (https://www.novogene.com/, accessed on 21 October 2021). The raw data of sequencing were subject to sequence read archive database (BioProject accession number: PRJNA902675).

The raw sequencing data were processed using SMRT Link v7.0 software. The circular consensus sequences (CCSs) were obtained from Subread (parameters: min_length = 50, max_length = 15,000, min passes = 1). Full-length non-chimeric (FLNC) reads were recognized by checking the polyA tail and cDNA adaptors. The polished consensus sequences were obtained from the FLNC reads using an isoform-level clustering algorithm and Quiver [30]. Next, the polished consensus sequences were corrected with the Illumina RNA-seq data (unpublished data) using LoRDEC v0.7 [31]. Finally, redundancy was removed from the corrected consensus sequences using CD-HIT (parameter: -c 0.95) [32], and the full-length transcripts were used for subsequent analysis.

### 2.3. Unigene Functional Annotation and Structure Prediction

To obtain comprehensive gene functional annotations, all the unigenes were subjected to BLAST against non-redundant protein sequences (NR), nucleotide sequences (NT), protein families (Pfam), Swiss–Prot, Kyoto Encyclopedia of Genes and Genomes (KEGG), Gene Ontology (GO), and Eukaryotic Ortholog Groups (KOG) databases with an E-value ≤ 1 × 10^−5^. In addition, the unigenes that are associated with the important functions that are described in this study were manually examined to verify the annotations as previously described [33].

The obtained unigenes were used for CDS prediction with ANGEL software (https://github.com/PacificBiosciences/ANGEL, accessed on 12 March 2022). The ITAK software was used to predict the *H*. *akashiwo* TFs. Then, simple sequence repeat (SSR) detection was performed with MISA (https://pgrc.ipk-gatersleben.de/misa/, accessed on 12 March 2022). In addition, four programs including Coding-Non-Coding-Index (CNCI), Coding Potential Calculator (CPC), Pflam, and PLEK were used to predict the lncRNAs.

## 3. Results

### 3.1. Single-Molecule Real-Time Sequencing Profiling

The polymerase read of the full-length transcriptome of *H*. *akashiwo* was 48.66 Gb, and 45.44 Gb of subread bases were obtained after the initial quality control. After transcript merging, a total of 706,413 CCSs were generated with an average length of 1828 bp. Furthermore, 69.86% (493,483) of the CCSs were identified as FLNC, and 46,954 polished consensus sequences were generated. Finally, the obtained transcripts were clustered, and 16,668 unigenes were generated, with an N50 length of 2007 bp (Table 1 and Table 2 and Appendix A).

### 3.2. Unigene Function Annotation

To obtain comprehensive information on the 16,668 unigene functions, the unigenes were annotated using seven databases, as specified above (Appendix A). As shown in Figure 1, a total of 8666 (52.00%) unigenes were successfully annotated using at least one database. Of these, 6558 (39.34%) and 5894 (35.36%) were annotated using NR and Swiss–Prot, respectively.

The GO analysis showed that 6478 (38.86%) unigenes were classified into three major categories, including biological processes, cellular components, and molecular functions. In terms of biological processes, cellular and metabolic processes were the most enriched groups. Within molecular functions, binding and catalytic activity accounted for the largest proportion. Then, under cellular components, cell and cell parts were the most abundant groups (Figure 2).

For the KEGG classification, 6551 (39.30%) unigenes were annotated to different functional groups, including cellular processes, organismal systems, and metabolism. Among them, the unigenes that are involved in metabolism account for 29.57%. In addition, there were 416 unigenes associated with energy metabolism (Figure 3).

### 3.3. Genes Related to Phosphorus and Nitrogen Metabolism

Based on the results of the gene annotation using the seven databases, we also explored genes that are related to N and P uptake and metabolism. In the generated full-length transcriptome of *H*. *akashiwo*, the genes that are related to N uptake were detected, including high-affinity nitrate transporter 2.5 (*NRT*), ammonium transporter (*AMT*), urea-proton symporter degradation of urea 3 (*DUR3*), and amino acid transporter. Most of the genes encoding proteins that are required for N metabolism were also identified. For instance, genes that encode nitrate reductase (NR) and nitrite reductase were identified, which are involved in nitrate assimilation. Additionally, almost all the urea cycle-related genes, including carbamoyl-phosphate synthase and ornithine transcarbamylase, were detected (Figure 4 and Appendix A). To cope with P stress, phytoplankton have developed a range of strategies. The genes encoding sodium-dependent phosphate transporter (SPT) and phosphate-repressible phosphate permease pho-4 (PHO4) were detected. Furthermore, several genes, including sulfate transporter and adenosine 5′-phosphosulfate reductase, which are involved in the regulation of sulfur assimilation and sulfolipid biosynthesis, were also detected (Figure 5 and Appendix A).

### 3.4. Analyses of Coding Sequences, Transcription Factors, Long Non-Coding RNAs, and Simple Sequence Repeats

A total of 14,824 CDSs were detected, which accounted for about 88.94% of the obtained unigenes. To validate the reliability of the SMRT sequencing, the predicted amino acid sequences of the four published blue light receptor aureochromes (*Haauro1*, *Haauro2*, *Haauro3*, and *Haauro4*) were also aligned against the CDSs of *H*. *akashiwo* using the local BLAST program (http://www.ncbi.nlm.nih.gov/blast/Blast.cgi, accessed on 15 March 2022) [37]. Two of them (*Haauro1* and *Haauro3*) had 99% identity, both with only one amino acid difference (Appendix A). A total of 300 putative TF members were found, which were categorized into 29 transcript families. Among them, the top TFs (HSF, C3H, TRAF, zn-clus, and mTERF) accounted for 43.33% (Figure 6a and Appendix A). In addition, there were 6203, 7378, 10221, and 7324 putative lncRNAs identified by CNCI, CPC, Pflam, and PLEK, respectively. Moreover, a total of 4392 putative lncRNA transcripts were predicted in *H*. *akashiwo* by four computational methods (Figure 6b and Appendix A). For suitable molecular development, a total of 7851 SSR markers were identified by the MISA program. Among them, 749 (9.54%) and 2595 (33.05%) were di-nucleotide and tri-nucleotide repeats, respectively (Figure 7 and Appendix A). These SSR markers could be considered for clarifying the population structure of *H*. *akashiwo* as was previously reported [38].

## 4. Discussion

Previous studies showed that several biological features of *H*. *akashiwo*, including encystation formation and diel vertical migration, are probably related to its bloom formation [12,18]. To explore the molecular mechanisms of these features, in several studies, targeted approaches, such as gene cloning, quantitative real-time PCR, and Illumina sequencing have been used [13,14,24,39]. However, molecular studies on HAB species have evolved slowly. This may be partly due to the lack of a reliable reference genome or well-annotated transcriptome. As with most protists, whole genome studies of HAB species, including *H*. *akashiwo*, have lagged behind model organisms (e.g., the diatom *Thalassiosira pseudonana* and green alga *Chlamydomonas reinhardtii*), due to their massive sized genomes (e.g., dinoflagellates) [22,40,41]. For instance, it has been reported that the *H*. *akashiwo* (formerly *Olisthodiscus luteus*) has an unclear genome size of around 2.9 Gb [42]. In this study, SMRT sequencing was conducted to generate the first full-length transcriptome of *H*. *akashiwo*. In total, 46,954 high-quality polished consensus sequences were obtained. Additionally, 16,668 unigenes were generated after removing redundancy. When compared with the previously published transcriptome of *H*. *akashiwo*, the number of unigenes that were obtained by SMRT sequencing was low (Table 2). However, the N50 length of the SMRT sequencing was substantially higher than the transcriptome of *H*. *akashiwo* that was generated by Illumina sequencing. This result indicates that SMRT sequencing is a suitable method for obtaining a reference transcriptome of *H*. *akashiwo*.

To obtain more detailed information about the target species transcriptome, the strategy of pooling samples from different treatments or tissues has been frequently performed with SMRT sequencing [25,28]. For instance, three different growth stages and several nutritional culture conditions of the dinoflagellate *Akashiwo sanguinea* were collected for SMRT sequencing library construction [25]. For *H*. *akashiwo*, previous studies showed that the light–dark cycle is an important environmental signal in the regulation of this species’ many biological processes [29]. For instance, 2826 differentially expressed genes (DEGs) were identified between two bloom samples (afternoon vs. morning). In contrast, only 304 DEGs were identified between bloom and pre-bloom samples, which were collected in the morning [29]. Thus, sample collection over a diel cycle (four representative time points) was used for cDNA library preparation in this study. Among the 16,668 obtained unigenes, 52.00% of the unigenes were annotated by at least one database. The annotation rate in this study is higher than that of the do novo transcriptome assembly of *H*. *akashiswo* (approximately 37%) but it is comparable to those of other bloom-forming algae [13,43,44]. These results also highlight the difficulty in annotating the transcriptome of non-model organisms [40]. Among 8666 annotated unigenes, 6478 (38.86%) and 6551 (39.30%) were enriched in the GO terms and the KEGG database, respectively. It provided a large amount of useful genetic information that could be used for gene functional studies and understanding the mechanism of *H*. *akashiwo* bloom formation at the molecular level.

To cope with N and P deficiency and utilize their different forms, *H*. *akashiwo* has evolved various strategies, such as increasing N or P transporters to enhance nutrient uptake [13,24]. In the generated transcriptome data, the major genes that are involved in N and P metabolism were detected. For instance, several genes (e.g., *NRT*, *AMT*, and *DUR3*) related to N substrate uptake were detected. Increasing mRNA abundance of these N transporters will benefit *H*. *akashiwo* in absorbing exogenous N [14]. In addition, NR is key enzyme of nitrate assimilation. Coyne et al. (2010) found that the transcript level of NR can change rapidly in response to N status variation in the environment [39]. Correspondingly, *SPT* and *PHO4*, which facilitate the diffusion of phosphate, were detected. In cyanobacteria and eukaryotic algae, a common response to low P stress is the up-regulation of P transport systems. Haley et al. (2017) reported that mRNA levels of multiple *H*. *akashiwo* P transporters, including PHO4-like, were markedly up-regulated in low P treatment [24]. In many phytoplankton, alkaline phosphatase (AP) plays an important role in the hydrolysis of dissolved organic P, and they are typically increased in abundance in low conditions. In laboratory and field studies, the mRNA abundance of AP was markedly up-regulated in *H*. *akashiwo* under conditions of P-depletion [13,24,29]. However, we failed to detect the transcript of AP, indicating that this gene may be expressed at low levels in our collected samples. In addition, we also did not detect two blue light receptors, *Haauro2* and *Haauro3*, although the other two genes (*Haauro2* and *Haauro4*) were detected with complete CDSs. These results further illustrate that collecting multiple types of samples for library construction is one of the key factors in obtaining a general encyclopedia of gene transcriptions. Thus, this issue needs careful consideration for future studies.

Long non-coding RNA belongs to the transcriptional class, and it was first described during the large-scale sequencing of full-length cDNA libraries of mice [45]. Previous studies have shown that the functions of lncRNAs are complex, including the regulation of gene transcription and organization of subcellular structures [46]. In this study, 4392 lncRNAs were detected in 16,668 transcripts. However, the function of these lncRNAs, including their potential target genes, still needs further experimental verification. In living organisms, TFs are essential for the regulation of gene expression. In photosynthetic stramenopiles, the diversity of the TFs was identified, and gene expression analysis showed that many of the TFs identified were probably involved in responses to particular stimuli [47]. For instance, 212 and 258 TFs were detected in *Phaeodactylum tricornutum* and *T*. *pseudonana*, respectively. In this study, we identified 300 putative TFs in *H*. *akashiwo*, accounting for 1.55% of their transcripts. Interestingly, a high abundance of HSFs were found in *H*. *akashiwo*, which is consistent with previous findings in two diatoms [47]. Moreover, 11 basic leucine zipper TFs, including two blue photoreceptors (*Haaureo1* and *Haaureo3*), were also identified in *H*. *akashiwo*. It has been previously shown that *Haaureo1* mRNA abundance exhibits a clear diel rhythm, and probably regulates the cell division cycle in *H*. *akashiwo* [37].

## 5. Conclusions

In this study, a relatively high-quality and complete reference transcriptome of *H*. *akashiwo* was obtained using SMRT sequencing. Furthermore, gene functional annotation, TFs, lncRNAs, and SSR analysis were investigated. There are 16,668 unigenes were obtained, 8666 (52.00%) were annotated in the seven public database. In addition, the majority of genes involved in N and P metabolism were identified. A total of 4392 lncRNAs, 300 putative TFs, 7851 SSR markers were predicted, respectively. These results provide a valuable resource for understanding the eco-physiological features of *H*. *akashiwo* (e.g., four generation mechanisms mentioned above) at a molecular level. Moreover, it is important to collect multiple types of samples for library construction. In future research, we should pay attention to this issue.

## Figures and Tables

**Figure 1 microorganisms-11-00389-f001:**
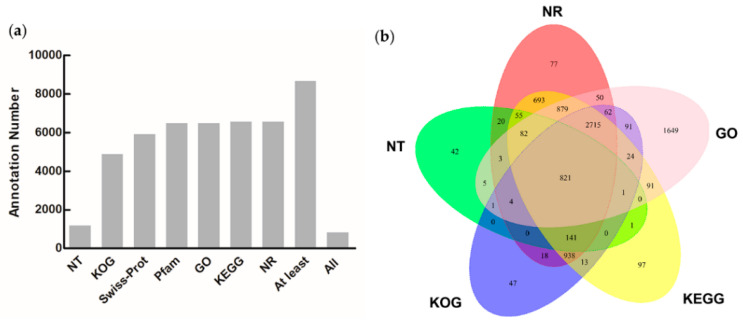
Functional annotation of the *Heterosigma akashiwo* full-length transcriptome. (**a**) Summary of the unigene annotation with seven public databases. At least: the number of unigenes that were annotated with at least one database. All: the number of unigenes that were annotated with all the databases; (**b**) Venn diagram of the annotation among the five databases.

**Figure 2 microorganisms-11-00389-f002:**
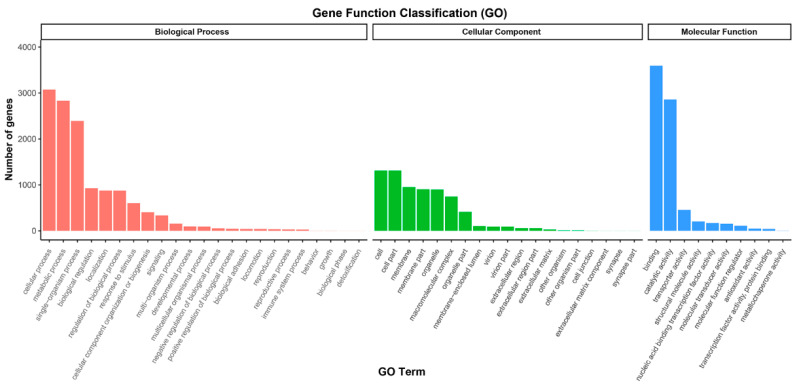
Gene Ontology functional classification of all the unigenes of *Heterosigma akashiwo*.

**Figure 3 microorganisms-11-00389-f003:**
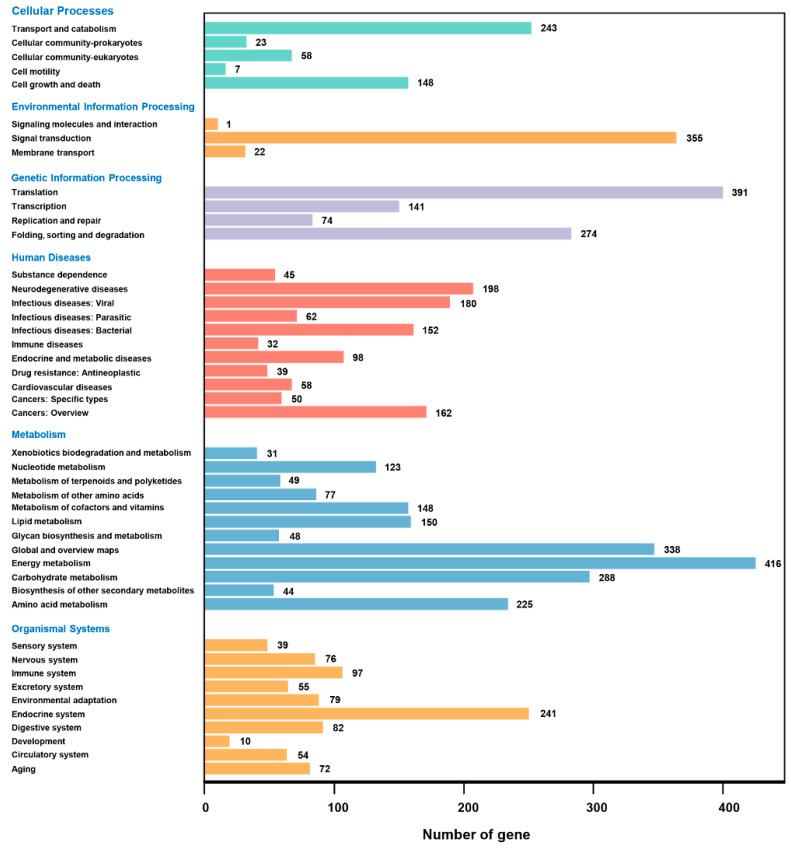
Kyoto Encyclopedia of Genes and Genomes classification of all the unigenes of *Heterosigma akashiwo*.

**Figure 4 microorganisms-11-00389-f004:**
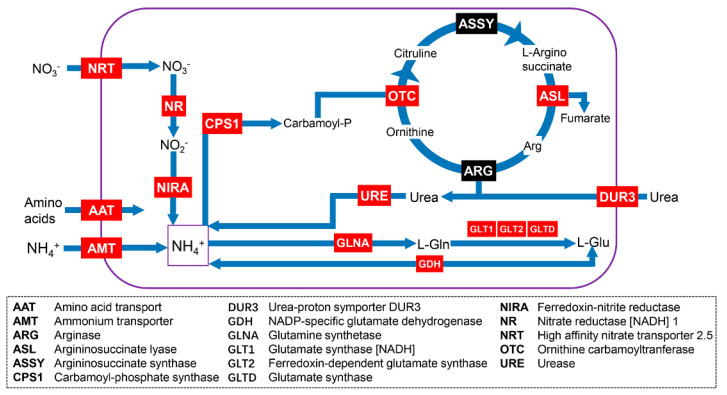
A cell model showing the unigenes of *Heterosigma akashiwo* involved in nitrogen uptake and metabolism. This model was modified from previous studies [34]. The red and black boxes represent the unigenes that were detected and not detected, respectively. The detail information of these genes is listed in Appendix A.

**Figure 5 microorganisms-11-00389-f005:**
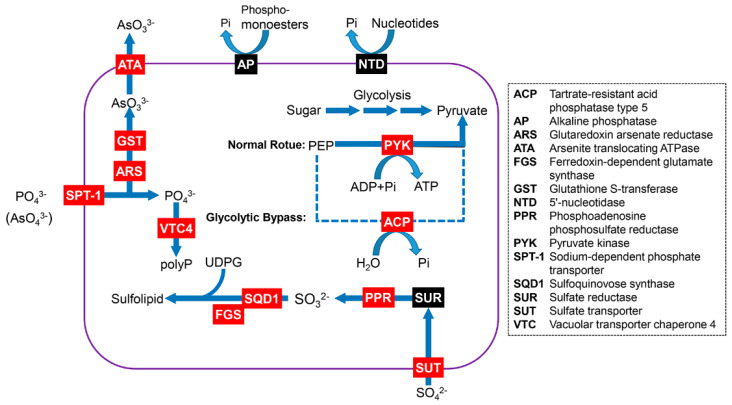
A cell model showing the unigenes of *Heterosigma akashiwo* that are involved in phosphorus uptake and metabolism. This model was modified from previous studies [35,36]. The red and black boxes represent the unigenes that were detected and not detected, respectively. The detail information of these genes is listed in Appendix A.

**Figure 6 microorganisms-11-00389-f006:**
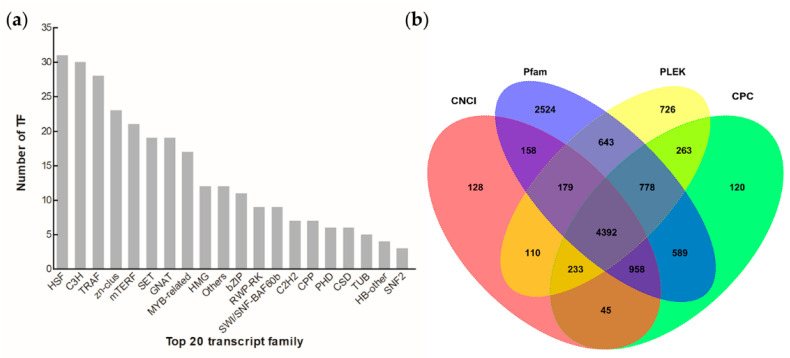
Identification of the transcription factors (TFs) and long non-coding RNAs (lncRNAs) of *Heterosigma akashiwo*. (**a**) The distribution of the top 20 transcript families. (**b**) A Venn diagram of the lncRNAs that were predicted by the CNCI, CPC, Pflam, and PLEK. The detail information of these genes is listed in Appendix A.

**Figure 7 microorganisms-11-00389-f007:**
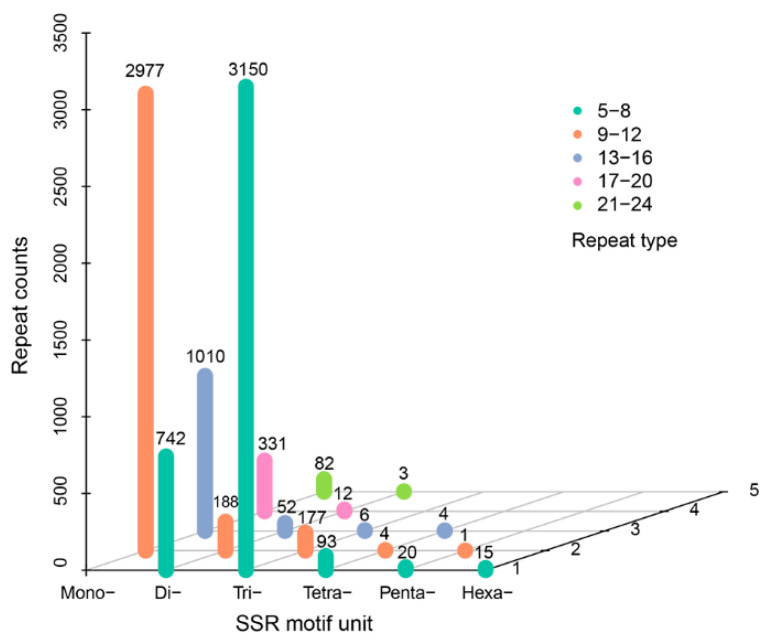
The distribution of the detected simple sequence repeats (SSRs) in the *Heterosigma akashiwo* transcriptome. Mono: mononucleotide; Di: dinucleotide; Tri: trinucleotide; Tetra: tetranucleotide; Penta: pentanucleotide; and Hexa: hexanucleotide. The detail information of these genes is listed in Appendix A.

**Table 1 microorganisms-11-00389-t001:** Summary of the *Heterosigma akashiwo* single-molecule real-time sequencing data.

Polymerase Read Bases (G)	Subread Bases (G)	FLNC ^a^ Number	Polished Consensus Sequences	Unigenes/N50 (bp)
48.66	45.44	493,483	46,954	16,668/2007

^a^: full-length non-chimeric.

**Table 2 microorganisms-11-00389-t002:** Information on the available *Heterosigma akashiwo* transcriptome.

Strain	SP ^a^	Unigenes/CDS ^b^	AM ^c^	N50 (bp)	References
CCMA369	PacBio Sequel	16,668	-	2007	This study
CCMA369	Illumina	108,924	Trinity	1015	[13]
CCMP2393	Illumina	40,801	BPA and ABySS	1402	[24]
CCMP3107	Illumina	18,721	BPA and ABySS	-	[23]
CCMP452	Illumina	17,723	BPA and ABySS	-	[23]
NB	Illumina	32,905	BPA and ABySS	-	[23]

^a^: Sequencing platform; ^b^: Coding sequence; ^c^: Assembly method; BPA: Bath Parallel Assembly.

## Data Availability

In this study, all data generated are included in this article and Appendix A files of it. Further enquiries can be directed to the corresponding author.

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
