# Peer review of "Full-Length Transcriptome Analysis of the Ichthyotoxic Harmful Alga Heterosigma akashiwo (Raphidophyceae) Using Single-Molecule Real-Time Sequencing"

_microorganisms, 2023, doi:10.3390/microorganisms11020389_

Round 1

Reviewer 1 Report

The present manuscript reports a full transcriptomic analysis for coding and non-coding RNAs to the raphidophyte Heterosigma akashiwo resulted to affected phosphorus (P) and N metabolism. This is an interesting and well-written work, but there are some issues that need to be addressed.

Some Major comments and suggestions are listed below:

1.     The authors should identify some of the deregulated genes related with phosphorus and nitrogen metabolism via qPCR.

2.     Can the authors comment on the up or down regulation of genes related with the phosphorus or nitrogen metabolism especially between the GM1-GM4 generation mechanisms?

3.     Except of the phosphorus and nitrogen related genes, the authors should provide a differential expression analysis for identifying the 10-20 most up or down regulated genes at 10, 14, 20h compared to D0 and try to connect these deregulated genes with the major affected pathway at the Heterosigma akashiwo.

Author Response

General Comments

The present manuscript reports a full transcriptomic analysis for coding and non-coding RNAs to the raphidophyte Heterosigma akashiwo resulted to affected phosphorus (P) and N metabolism. This is an interesting and well-written work, but there are some issues that need to be addressed.

Response: Thanks very much for your positive comments. We carefully revised the manuscript to the comments provided by you and the other reviewer. In this study, a full-length transcriptome of H. akashiwo was generated using SMRT sequencing. The obtained database provides a full-length transcriptome for the study of the ecology and physiology of H. akashiwo in future. We realized that it is essential to acknowledge there are much more work is needed to do in the future. For instance, identification of differentially expressed genes under different culture conditions still needs future study. Our detailed responses to all your comments are shown below.

Specific comments:

Comment 1: The authors should identify some of the deregulated genes related with phosphorus and nitrogen metabolism via qPCR.

Response: I think there is a misunderstanding here. In this study, a high-quality reference transcriptome for harmful alga Heterosigma akashiwo was obtained. The obtained reference transcriptome will be useful for future gene expression studied on H. akashiwo. As described in the revised manuscript, this study provides a valuable reference transcriptome for understanding how H. akashiwo blooms at a molecular level.

Comment 2: Can the authors comment on the up or down regulation of genes related with the phosphorus or nitrogen metabolism especially between the GM1-GM4 generation mechanisms?

Response: We appreciated the proposed experiments and discussion. I feel there may be a misunderstanding: we are not conducting differential gene expression analysis under different culture conditions but rather collection of diel samples for PacBio sequencing library construction. This study will lay a strong foundation for future studies that address GM1-GM4 generations in greater detail. In the revised manuscript, the relevant information have been added. We proposed that this study provide a valuable resource for understanding the eco-physiological features of H. akashiwo (e.g. four generation mechanisms mentioned above) at a molecular level. Please see line 312-314.

Comment 3: Except of the phosphorus and nitrogen related genes, the authors should provide a differential expression analysis for identifying the 10-20 most up or down regulated genes at 10, 14, 20h compared to D0 and try to connect these deregulated genes with the major affected pathway at the Heterosigma akashiwo.

Response: This is a very instructive comment. And, this will be certainly interesting to address in more detail in another future research project. In the manuscript, we proposed that the polished consensus sequences were corrected with the Illumina RNA-seq data (unpublished data) using LoRDEC v0.7. Thus, generation of a full-length transcriptome of H. akashiwo is our current research focus. In an ongoing project (unpublished data), we are investigating diel gene expression of H. akashiwo using the obtained full-length transcriptome and Marine Microbial Eukaryotic Transcriptome Sequencing Program database. To characterize diurnal gene expression in H. akashiwo, a total of 27 samples (three biological replicates per sample) were collected over diel cycle (Figure 1, for review purpose, unpublished data). Over 54% of measured transcriptome was regulated by photoperiod (Figure 2, for review purpose, unpublished data). Thanks again for this insightful comment.

Reviewer 2 Report

Manuscript entitled “Full-length transcriptome analysis of the ichthyotoxic harmful alga Heterosigma akashiwo (Raphidophyceae) using single-molecule real-time sequencing” submitted by Nanjing Ji, Xueyao Yin, Yujiao Chen, Yifan Chen, Mingyang Xu, Jinwang Huang, Yuefeng Cai and Xin Shen, can be considered for publication in Microorganisms Journal, after a major revision.

Here is a list of my specific comments:

  1. Page 1, line 30: “The raphidophyte Heterosigma akashiwo is a typical…”. This is a micro-or macro-algae??? This should be mentioned here.
  2. Page 1, line 35: “and CO2 level”. Correct the chemical formula of carbon dioxide. Make this correction on entire manuscript.
  3. Page 2, line 69: “To the best of our knowledge…”. This paragraph should be reworded, or deleted.
  4. Page 8, 4. Discussion: All experimental results presented in the previous section should be clearly and detailed discussed here. Pay attention on the interpretation of these results in accordance with the main objectives of this study.
  5. Page 8, line 264: “To cope with N and P deficiency and utilize their different forms…”. These observations should be more detailed discussed. How was proved the existence of these two strategies???
  6. Page 9, line 285: “In photosynthetic stramenopiles…”. Why it is important this aspect??? Please explain.
  7. Page 9, 5. Conclusions: This section is too brief and should be detailed. Include in this section the most important experimental results and findings to highlight the importance of this study.

Author Response

General Comments

Manuscript entitled “Full-length transcriptome analysis of the ichthyotoxic harmful alga Heterosigma akashiwo (Raphidophyceae) using single-molecule real-time sequencing” submitted by Nanjing Ji, Xueyao Yin, Yujiao Chen, Yifan Chen, Mingyang Xu, Jinwang Huang, Yuefeng Cai and Xin Shen, can be considered for publication in Microorganisms Journal, after a major revision.

Response: We appreciate your valuable comments. We carefully revised the manuscript according to the comments provided by you and the other reviewer. First, we added information on marine microalgae H. akashiwo. Second, the confused sentences have been rewritten. Third, the discussion and conclusions have been enriched. Our detailed responses to all your comments are shown below. Please check.

Specific comments:

Comment 1: Page 1, line 30: “The raphidophyte Heterosigma akashiwo is a typical…”. This is a micro-or macro-algae? This should be mentioned here.

Response: Thanks for your suggestion. The raphidophyte Heterosigma akashiwo is marine microalgae. This sentence has been rewritten in the revised manuscript. Please see line 31.

Comment 2: Page 1, line 35: “and CO2 level”. Correct the chemical formula of carbon dioxide. Make this correction on entire manuscript.

Response: Thanks for catching this mistake. In the revised manuscript, the chemical formula of carbon dioxide has been corrected. Please see line 35 and 62.

Comment 3: Page 2, line 69: “To the best of our knowledge…”. This paragraph should be reworded, or deleted.

Response: This sentence has been deleted in the revised manuscript. Please see line 68.

Comment 4: Page 8, 4. Discussion: All experimental results presented in the previous section should be clearly and detailed discussed here. Pay attention on the interpretation of these results in accordance with the main objectives of this study.

Response: We have enriched the discussion following your suggestions. For instance, we proposed that obtained database provided a large amount of useful genetic information for functional investigation in H. akashiwo. Please check the revised manuscript.

Comment 5: Page 8, line 264: “To cope with N and P deficiency and utilize their different forms…”. These observations should be more detailed discussed. How was proved the existence of these two strategies???

Response: In the revised manuscript, the discussion in this section has been enriched, and relevant references were added. Please see line 266-278.

Comment 6: Page 9, line 285: “In photosynthetic stramenopiles…”. Why it is important this aspect??? Please explain.

Response: Thanks for your suggestion. In photosynthetic stramenopiles, the diversity of the TFs was identified, expression data analysis shows that many of the TFs studied are transcribed and may be involved in specific responses to environmental stimuli. In the revised manuscript, this part has been rewritten. Please see 294-297.

Comment 7: Page 9, 5. Conclusions: This section is too brief and should be detailed. Include in this section the most important experimental results and findings to highlight the importance of this study.

Response: In the revised manuscript, the most the most important experimental results have been added in the Conclusions. In this study, a relatively high-quality and complete reference transcriptome of H. akashiwo was obtained using SMRT sequencing. Furthermore, gene functional annotation, TFs, lncRNAs, and SSR analysis were investigated. There are 16,668 unigenes were obtained, 8666 (52.00%) were annotated in the five public database. In addition, the majority of genes involved in N and P metabolism were identified. A total of 4392 lncRNAs, 300 putative TFs, 7851 SSR markers were predicted, respectively. These results provide a valuable resource for understanding the eco-physiological features of H. akashiwo (e.g. four generation mechanisms mentioned above) at a molecular level. Moreover, it is important to collect multiple types of samples for library construction. In future research, we should pay attention to this issue. Please see line 307-316.

Round 2

Reviewer 1 Report

Dear authors,

Thank you for your responses to my comments. I would like to clarify that no misunderstanding took place. In general, any omics data deliver broad information and some of my comments could have been responded in a better way.

I can accept, for example, the fact that the authors need more data for the GM1-GM4 mechanisms, but not that they can not verify some of the deregulated genes with a PCR. The major problem is on my third comment. They should have information regarding the deregulated genes. They talk about phosphorus and nitrogen metabolism, but they did not show which of these genes are deregulated. 

Reviewer 2 Report

All my previous remarks and comments have been considered in this new version of the manuscript. In my opinion, the revised manuscript meets the criteria and can be published as original paper in Microorganisms Journal.